



# Tropical wet season runoff mobilises younger carbon in rainforest streams but older carbon in agricultural streams

Clément Duvert[1,2], Vanessa Solano[1,8], Dioni I. Cendón[4,5], Francesco Ulloa-Cedamanos[1], Liza K. McDonough[4,5], Robert G. M. Spencer[6], Niels C. Munksgaard[2], Lindsay B. Hutley[1], Jean-Sébastien Moquet[7], David E. Butman[3]

[1]Research Institute for the Environment and Livelihoods, Charles Darwin University, Darwin, NT, Australia

[2]College of Science and Engineering, James Cook University, Cairns, QLD, Australia

[3]School of Environmental and Forest Sciences, University of Washington, Seattle, WA, USA

[4]Australian Nuclear Science and Technology Organisation, Lucas Heights, NSW, Australia

[5]School of Biological, Earth and Environmental Sciences, UNSW Sydney, Sydney, NSW, Australia

[6]National High Magnetic Field Laboratory Geochemistry Group, Department of Earth, Ocean, and Atmospheric Science, Florida State University, Tallahassee, FL, USA

[7]Institut des Sciences de la Terre d'Orléans, Université d'Orléans-CNRS-BRGM, Orléans, France

[8]Now at : Groupe de Recherche Interuniversitaire en Limnologie, Département des Sciences Biologiques, Université du Québec à Montréal, Montréal, QC, Canada

*Correspondence to*: Clément Duvert (clem.duvert@cdu.edu.au)

**Abstract.** Knowledge of the age of organic carbon (C) that is leached from soils to streams is key to understanding how C is mobilised within ecosystems. The tropics are characterised by significant C fluxes through streams, yet the time scales of organic C sequestration and export remain uncertain in these regions. Here we examined the concentration, composition and age of dissolved organic C (DOC) in 18 small mountainous catchments of the Australian humid tropics, including six rainforest and 12 agricultural catchments, sampled during both the dry and wet seasons. We found that DOC ages varied widely across sites but were generally centuries to millennia old (median ± standard deviation 1,553 ± 848 years BP), with no consistent differences between rainforest and agricultural catchments. However, the two land use categories diverged in their responses to high flow conditions, with DOC age in rainforest streams tending to decrease from 1,878 ± 604 years BP in the dry season to 708 ± 791 years BP in the wet season, whereas agricultural streams mobilised similarly aged or older DOC in the wet season (1,728 ± 641 years BP) than in the dry season (1,303 ± 1,036 years BP). A subset of dissolved inorganic C (DIC) samples collected from three of the catchments (both rainforest and agricultural) indicated that DIC was mostly modern (123 ± 136 years BP) and always younger than DOC. These differences in DIC and DOC ages suggest a partial decoupling between DOC and DIC export pathways, with DOC derived from older soil C pools, while DIC reflected recent C inputs from vegetation uptake and decomposition. Our results highlight the importance of seasonal shifts in the age of C export and the need to conduct sampling that encompasses seasonality in human-impact studies to better constrain C pools and sinks.





## 1    Introduction

Soils represent one of the largest reservoirs of organic carbon (C) and play a crucial role in the global terrestrial C sink. Yet
the stability of the soil C pool is highly vulnerable to anthropogenic disturbance, including urban and agricultural development
(Don et al., 2011; Schlesinger, 1986; Xenopoulos et al., 2021; Hobley et al., 2017). Soil C loss not only occurs during the
process of land use change, when carbon dioxide ($CO_2$) is released to the atmosphere, but also with changes in the leakage of
organic C into streams and rivers as dissolved organic C (DOC) (e.g. Coble et al., 2022; Wilson and Xenopoulos, 2008; Drake
et al., 2020). While most riverine DOC is modern, sourced from recent assimilation of $CO_2$ by vegetation (e.g. Mayorga et al.,
2005; Marwick et al., 2015; Billett et al., 2007), land conversion to agriculture can cause old soil organic C to re-enter the
contemporary C cycle and be exported via streams and rivers (Butman et al., 2015; Evans et al., 2014; Moore et al., 2013).
Conversion from natural to agricultural land use has been linked to increased soil erosion and to the alteration of surface and
subsurface flow paths and water residence times following disturbance (Barnes et al., 2018; Butman et al., 2015; Evans et al.,
2022; Drake et al., 2020; Drake et al., 2019).

Tropical streams and rivers are a globally significant hotspot for terrestrial C export (Liu et al., 2022; Liu et al., 2024; Rocher-
Ros et al., 2023; Battin et al., 2023), yet knowledge of the origin and age of C export from tropical ecosystems remains limited.
Early work in the Amazon (Mayorga et al., 2005) and Congo (Spencer et al., 2012) revealed that the DOC transported by these
two major rivers was predominantly modern. Mayorga et al. (2005) went on to show that this young DOC was the dominant
source of dissolved inorganic C (DIC) and $CO_2$ degassing. Marwick et al. (2015) also consistently found modern DOC in a
range of African rivers. The dominance of young DOC has also been highlighted in smaller river systems, including streams
draining undisturbed peatland forest in Borneo (Moore et al., 2013; Müller et al., 2015) and pristine rainforest in the upper
Congo Basin (Drake et al., 2019). However, anthropogenic disturbance has emerged as a key control on DOC age in small
tropical systems. Studies have shown that in catchments where land has been cleared for agriculture, exported DOC can be
centuries to millennia old (Moyer et al., 2013; Drake et al., 2019; Moore et al., 2013; Waldron et al., 2019).

Despite improved understanding of the role of land use change on DOC age in tropical streams and rivers, observations remain
limited in both space and time, and the extent to which the age of riverine DOC is affected by changes in flow conditions
remains unknown. The tropics are affected by strong rainfall seasonality, and the bulk of C export is likely to occur during the
wetter months, yet few studies have examined DOC age under high flow conditions and across varying flow regimes. Marwick
et al. (2015) reported younger DOC during the wet season in the Athi River (Kenya), attributing this to the flushing of young
organic C accumulated during the dry season. Likewise, Chen et al. (2024) found younger DOC values during a large flood of
the subtropical Yangtze River compared to lower flow conditions. Similar patterns have been reported in northern high-latitude
and temperate regions, where high hydrological connectivity and shallow subsurface flow paths tend to mobilise younger DOC
(Tittel et al., 2022; Campeau et al., 2019; Barnes et al., 2018; Aiken et al., 2014). In contrast, Moore et al. (2013) found
increasing DOC ages with wet conditions in disturbed peatland catchments of Borneo. These contrasting findings highlight
the remaining uncertainties regarding the combined effects of land use and flow conditions on the age of transported DOC in





tropical streams. Also unknown is the relationship between DOC and DIC age in these systems. Studies have reported a decoupling between DOC and DIC ages, with generally modern DIC/$CO_2$ compared to older and more variable DOC (Campeau et al., 2019; Moyer et al., 2013). Such patterns have been attributed to DIC/$CO_2$ being directly derived from recent soil $CO_2$ inputs from root respiration rather than from the decomposition of older DOC (Campeau et al., 2019), a hypothesis that needs to be further tested.

The humid tropics of Australia are a mountainous region that drains to the Great Barrier Reef lagoon. Agricultural development in the region began in the 1870s, with rainforest cleared for sugarcane and pasture development (Kemp et al., 2007). As a result, coastal ecosystems have been impacted by poor water quality, which has intensified in the past few decades (Kroon et al., 2016; Kroon et al., 2012; Guo et al., 2025). While suspended sediment and nutrient loads have been particularly scrutinised, few studies have investigated C export from rivers of the region (Rosentreter and Eyre, 2020), and none have investigated the age of exported C. In this paper, we aim to assess the impact of land disturbance and flow conditions on the age of DOC exported by streams across the Australian humid tropics. We examined the characteristics, concentration and isotopic composition (carbon-13; $\delta^{13}C$ and radiocarbon; $^{14}C$) of bulk DOC in 18 small catchments (six relatively pristine rainforest and 12 dominated by agriculture) during both the dry and wet seasons. Additionally, we measured the age of soil organic C in two shallow soil cores and the age of DIC in a small subset (three) of the catchments. We addressed the following questions:

(1) Is there a significant difference between the age of DOC transported in streams draining rainforest catchments versus that of agricultural catchments? Here we hypothesised that natural land conversion to agricultural results in the export of older DOC, as increased erosion results in the depletion of modern soil organic C.

(2) Does the age of DOC transported in streams alter with changing flow conditions? Here we hypothesised that DOC is younger at high flow than under baseflow conditions, as C from shallower soil layers tends to be mobilised when water tables rise.

(3) Is the age of DIC comparable to the age of DOC? Here we hypothesised that DIC is younger than DOC, due to external inputs of young soil $CO_2$ and to younger, more biolabile organic matter being preferentially mineralised.

## 2 Methods

### 2.1 Study area

The 18 stream sites are located within the humid tropics of Far North Queensland, Australia (Figure 1; Table 1). The region spans a range of climate types, from *Af* (tropical rainforest) and *Am* (tropical monsoon) in the lowlands (0–500 masl) to *Cfa* (humid subtropical) at higher elevations (500–1,000 masl) (Beck et al., 2023). Annual rainfall in the region ranges between approximately 1,500 and 4,000 mm and follows a strong seasonal pattern, with 65–75% falling during the wetter months (November to April) and 25–35% during the drier months (May to October) (Bureau of Meteorology, 2025). The geology of the region is diverse, with the coastal areas to the East dominated by Quaternary alluvial deposits, while the mountain range



and Atherton Tableland to the West are characterised by Cenozoic and Palaeozoic basaltic lava flows and metamorphic rocks. The area does not feature any significant organic-rich or carbonate-rich sedimentary formations (Jell, 2013).

**Figure 1: Location of the 18 study catchments. Catchments delineated in green represent rainforest (R) while those shaded in yellow to red represent agricultural catchments, dominated by either pasture (P) or cropland (C). Grey lines denote elevation contours. The background imagery is derived from Sentinel-2 satellite data.**

Samples were collected in six streams that drain relatively pristine rainforest remnants (R1 to R6) and 12 streams that drain some degree of agriculturally disturbed land, including six with land converted to pasture (P1 to P6) and six with cropland (mostly sugarcane and banana plantations; C1 to C6) (Figure 1). Some of the agricultural sites drain a mixture of different land uses (i.e., forested area, agricultural area (pasture/crops) and urban area; Table 1). However, urban land use remains below 4% at all sites. The six rainforest sites are generally located in more rugged terrain, resulting in higher mean catchment slopes (0.25 m/m) compared to the 12 disturbed sites (0.11 m/m) (p=0.005; Mann-Whitney U-test). Streams range from second to





fourth order, and while the extent of groundwater connection is unknown, it is likely that all streams receive some level of
shallow groundwater inputs.

**Table 1. Landscape features for the 18 stream sites. Elevations are reported as mean [min-max]. Infr. = infrastructure/urban. Site coordinates are available in Table S4.**

| site ID | stream name | stream order | area (km²) | elevation (masl) | slope (m/m) | forest (%) | pasture (%) | crop (%) | infr. (%) |
|---------|-------------|--------------|-----------|------------------|-------------|------------|-------------|----------|-----------|
| C1 | Victory Creek | 3 | 17.7 | 22 [3-114] | 0.042 | 19 | 0 | 77 | 4 |
| C2 | Mullins Rd Creek | 2 | 6.8 | 22 [11-172] | 0.043 | 20 | 1 | 79 | 0 |
| C3 | Scheu Creek | 3 | 5.2 | 43 [16-85] | 0.043 | 4 | 24 | 70 | 2 |
| C4 | Utchee Creek | 4 | 8.5 | 182 [72-503] | 0.188 | 61 | 5 | 34 | 1 |
| C5 | Mistake Creek | 3 | 8.5 | 25 [15-38] | 0.007 | 3 | 0 | 96 | 1 |
| C6 | Diggers Creek | 4 | 38.6 | 150 [13-826] | 0.183 | 71 | 0 | 27 | 2 |
| P1 | Short Creek | 3 | 6.7 | 780 [738-885] | 0.058 | 18 | 52 | 28 | 1 |
| P2 | Theresa Creek | 2 | 6.6 | 817 [757-947] | 0.120 | 21 | 78 | 0 | 2 |
| P3 | Tranter Creek | 3 | 10.0 | 925 [749-1372] | 0.151 | 40 | 60 | 0 | 0 |
| P4 | Wadda Creek | 4 | 9.8 | 268 [178-352] | 0.105 | 10 | 64 | 25 | 0 |
| P5 | Gregory Creek | 3 | 8.4 | 239 [150-326] | 0.087 | 14 | 78 | 6 | 2 |
| P6 | Mena Creek | 4 | 13.2 | 156 [67-484] | 0.185 | 56 | 36 | 8 | 1 |
| R1 | Kauri Creek | 3 | 16.4 | 948 [687-1264] | 0.296 | 100 | 0 | 0 | 0 |
| R2 | Boulders Creek | 3 | 17.5 | 345 [53-917] | 0.344 | 99 | 0 | 1 | 0 |
| R3 | Gooligan Creek | 2 | 4.8 | 467 [364-467] | 0.156 | 100 | 0 | 0 | 0 |
| R4 | Douglas Creek | 2 | 1.7 | 584 [512-694] | 0.134 | 100 | 0 | 0 | 0 |
| R5 | Bridge 15 Creek | 3 | 11.0 | 427 [57-929] | 0.261 | 100 | 0 | 0 | 0 |
| R6 | Coochable Creek | 3 | 6.2 | 589 [133-942] | 0.280 | 100 | 0 | 0 | 0 |

## 2.2  Sample collection

Stream water samples were collected once during the dry season from August 25th to 28th, 2020, and a second time at the end
of the wet season from March 21st to 26th, 2021 (Figure S1). At each site, electrical conductivity and temperature were
measured using a portable conductivity meter (WTW), while the partial pressure of $CO_2$ ($pCO_2$) was measured only during the
wet season campaign using a pre-calibrated sensor (eosGP, Eosense) that was allowed to equilibrate for at least 30 minutes
prior to measurement. At each site, samples were taken for later analysis of DOC concentration, $\delta^{13}$C-DOC, $^{14}$C-DOC,
dissolved organic matter (DOM) composition, and water stable isotopes (deuterium, $\delta^2$H and oxygen-18, $\delta^{18}$O). In addition, at
three sites (i.e. R1, P1 and C4; randomly selected from each land use type), samples were collected for $^{14}$C-DIC analyses.
We collected samples in 1 L high density polyethylene (HDPE) bottles ($^{14}$C-DIC), 2 L HDPE bottles ($^{14}$C-DOC), 125 mL
HDPE bottles (DOM composition), 40 mL pre-acidified borosilicate amber glass vials (DOC and $\delta^{13}$C-DOC), and 60 mL
polyethylene centrifuge vials ($\delta^2$H and $\delta^{18}$O). All HDPE bottles were acid-leached with hydrochloric acid and Milli-Q rinsed
prior to sampling. Samples for $^{14}$C-DIC, $^{14}$C-DOC, and DOM composition analyses were filtered in the field using a peristaltic



pump (Geopump, GEOTECH) connected to Tygon tubing and high-capacity 0.45 μm polyethersulfone filters (FHT-45, Waterra). The Tygon tubing was pre-rinsed in 10% hydrochloric acid and Milli-Q water between samples, and both the tubing and the high-capacity filters were then pre-conditioned with the water to be sampled for at least 30 seconds before sampling.

We also extracted two shallow soil cores in two rainforest sites (R3 and R6) using a hand auger. We sampled soil organic C (SOC) at three depths along these cores (0–5 cm; 10–15 cm; 20–30 cm) for later analysis of [14]C-SOC. For each depth, approximately 100 g of soil were collected, double Ziploc bagged, and frozen upon return to the laboratory.

### 2.3   Laboratory analyses

Both [14]C-DOC and [14]C-DIC analyses were conducted at the Australian Nuclear Science and Technology Organisation (ANSTO) in Sydney (NSW, Australia). [14]C-DOC samples were acidified to pH 2, sparged with $N_2$ gas and neutralised to just

below pH 7, rotary evaporated to a concentrate, and freeze-dried to a powder. [14]C-DIC samples were acidified on a water line under vacuum, with $CO_2$ released using phosphoric acid. The liberated $CO_2$ was then captured in a vial using liquid nitrogen. Samples for [14]C-DOC and [14]C-DIC were then combusted using CuO, Ag and Cu wire. $CO_2$ was cryogenically purified before being converted to graphite, as outlined in Hua et al. (2001). The graphite was analysed by accelerator mass spectrometry (Wilcken et al., 2015). We report all [14]C data as normalised percent Modern Carbon (pMC), calculated according to Stuiver

and Polach (1977). Mean standard errors were ±0.41 pMC for [14]C-DOC and ±0.36 pMC for [14]C-DIC.

All soil samples were sent to the Radiocarbon Laboratory at the Australian National University (Canberra, ACT) for [14]C-SOC analysis. Prior to analysis, rootlet fragments were sieved out and samples were pretreated with hydrochloric acid to remove any carbonates. The analyses were conducted on a single stage accelerator mass spectrometer following the method described in Fallon et al. (2010). Results are reported as pMC, calculated according to Stuiver and Polach (1977). The mean standard

error based on all six measurements was ±0.28 pMC.

DOM composition was analysed on an Aqualog optical spectrometer (HORIBA) at ANSTO, with excitation wavelengths of 240–600 nm and emission wavelengths of 250–800 nm. Data were blank corrected, interpolated, corrected for inner filter effects and for Raman scattering and first and second order Rayleigh scattering, and normalised using the Aqualog software (Hansen et al., 2018). We derived peak intensities following Coble (1996), and calculated the humification index (HIX) as per

Ohno (2002) and the biological index (BIX) as per Huguet et al. (2009) and Fellman et al. (2010). Peak intensities, HIX and BIX were determined using the R package *staRdom* (Pucher et al., 2019).

DOC concentrations and δ[13]C-DOC were measured at the Environmental Analysis Laboratory in Southern Cross University (Lismore, NSW) using a Flash elemental analyser (Thermo Fisher) connected to a Delta V Plus isotope ratio mass spectrometer (Thermo Fisher), as per Carvalho (2023). δ[2]H and δ[18]O were analysed at Charles Darwin University (Darwin, NT) on a L2130-

i cavity ring-down spectrometer (Picarro) connected to a diffusion sampler as described in Munksgaard et al. (2011).



## 2.4 Catchment delineation and land use classification

We conducted all spatial analyses in QGIS 3.4 (Qgis Development Team, 2019). Catchment boundaries and average catchment slopes were estimated using LiDAR-based elevation data at 5-m resolution (Geoscience Australia, 2015). In areas where the LiDAR data were incomplete or unavailable, we used elevation information from the SRTM 30 (Shuttle Radar Topography Mission, NASA). Land use proportions were derived from Sentinel-2 satellite imagery captured in October 2020. The classification between rainforest, pasture, cropland, and infrastructure was conducted using the Semi-Automatic Classification Plugin (Congedo, 2021) in QGIS 3.4.

## 2.5 Data analysis

There were very few differences between agricultural sites dominated by pasture and those dominated by cropland, with the only significant differences observed in elevation and stable isotopic values during the dry season (Table S1). Therefore, we grouped these two land uses into a single 'agricultural' category. To assess seasonal differences within the same category (rainforest or agricultural), we used the Wilcoxon signed-rank test (*signrank* function in MATLAB). To compare the two land use categories (rainforest and agricultural) within the same season (wet or dry), we used the Mann-Whitney U-test (*ranksum* function in MATLAB).

To explore the variability in $^{14}$C-DOC and identify potential drivers of these variations, we ran generalised additive models (GAMs) using the *fitrgam* function in MATLAB, as most relationships between $^{14}$C-DOC and predictors were nonlinear (Table S2). We developed separate GAMs for the dry and wet seasons as we expected the drivers to differ between our two sampling campaigns. All predictor variables were standardised (centred to the mean and scaled to the standard deviation) prior to analysis, and predictors that were strongly collinear were excluded. To assess collinearity, we calculated Pearson correlation coefficients and removed variables with coefficients greater than 0.75 (Figure S2). This led us to remove $\delta^{18}$O (strongly correlated with $\delta^{2}$H), elevation (strongly anticorrelated with $\delta^{2}$H and temperature), and the fraction of forest (strongly correlated with slope). Because slope significantly differed between the two land use categories (p=0.005; Mann-Whitney U-test), we added an interaction term ('land use * slope') to account for the non-independence between these two predictors. We assessed model performance by comparing Akaike Information Criterion (AIC) values, testing models with and without this interaction term to determine the best fit. The final set of predictors was as follows:

$^{14}$C-DOC ~ intercept + s(area) + s(temperature) + s(conductivity) + s(DOC) + s($\delta^{13}$C-DOC) + s($\delta^{2}$H) + s(BIX) + s(HIX) + s(land use * slope) + ε                                                                (1)

where s() indicates a smooth term and ε is the error term. We then used partial dependence plots to visualise the marginal effect of each predictor on $^{14}$C-DOC based on the fitted GAMs.

For the three sites where we collected $^{14}$C-DIC data, we used a two-endmember mixing approach to estimate the relative contributions of soil $CO_2$ inputs and DOC mineralisation to stream DIC. The expected absence of any significant geogenic DIC source in the study catchments (Jell, 2013) allowed us to undertake this analysis. To represent the soil $CO_2$ endmember



($C_{soil}$), we used the mean $^{14}$C-SOC value from the two shallowest SOC samples, i.e. 104.7 pMC. To characterise the DOC endmember ($C_{DOC}$), we used the $^{14}$C-DOC value measured at the site on the same day. The fraction of DOC mineralisation

($f_{DOC}$) to stream DIC ($C_{DIC}$) was then estimated as follows:

$$f_{DOC} = \frac{c_{DIC} - c_{soil}}{c_{DOC} - c_{soil}} \qquad (2)$$

This simple approach provided only coarse estimates of the sources of stream DIC, because $^{14}$C-DOC likely represents an older, less labile fraction of DOC, which could result in underestimations of $f_{DOC}$. However, the low HIX values (0.4–4.1) measured across all sites (see Results) indicate a predominance of highly biolabile DOM, suggesting that our mixing model

provided coarse yet reliable results.

## 3    Results

The two soil cores yielded contrasting results (Table S3), with modern SOC at all three depths at site R6 (all values > 104.6 pMC), whereas site R3 had modern (102.8 pMC) SOC only at the shallowest (0–5 cm) depth, and increasingly old SOC at greater depths (99.1 pMC at 10–15 cm and 83.4 pMC at 20–30 cm). All values above 100 pMC indicate samples affected

by the nuclear bomb pulse (e.g. Graven et al., 2020).

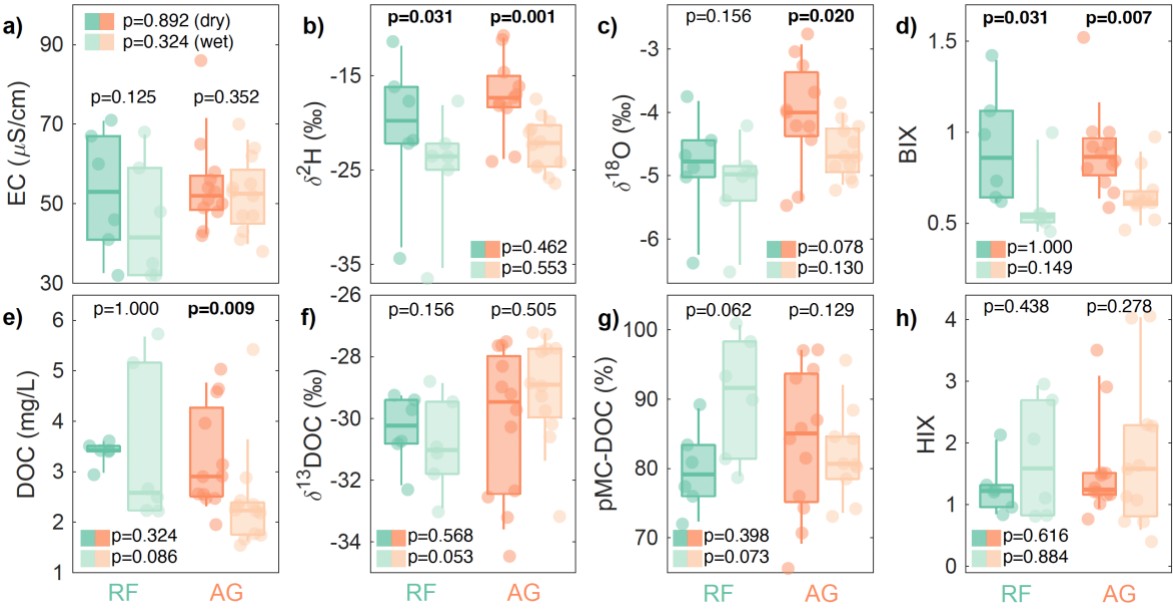

**Figure 2. Distribution of variables between the dry (dark) and wet (light) seasons across rainforest (RF; green) and agricultural (AG; orange) sites. Variables are a) electrical conductivity (EC), b) and c) water stable isotopes ($\delta^2$H and $\delta^{18}$O), d) biological index (BIX), e) DOC, f) $\delta^{13}$C-DOC, g) $^{14}$C-DOC in percent Modern C (pMC), and h) humification index (HIX). The p-values at the top of**

**each boxplot pair represent the seasonal differences within the same land use category based on Wilcoxon signed-rank tests (significant differences in bold). The p-values next to the coloured boxes represent the differences between land use categories for the dry (dark) and wet (light) seasons based on Mann-Whitney U-tests.**



We also observed large variations in measured stream water parameters across land use categories and seasons, although many
of these differences were not statistically significant (Figure 2). Water stable isotopes tended to decrease from dry (median -
17.7‰ for $\delta^2$H and -4.2‰ for $\delta^{18}$O) to wet seasons (median -22.9‰ for $\delta^2$H and -4.8‰ for $\delta^{18}$O), reflecting a change in rainfall
sources and reduced residence times in soils in the wet season, with less soil water subject to evaporation. This seasonal
decrease in $\delta^2$H and $\delta^{18}$O did not coincide with a decrease in EC (median from 52 to 50 µS/cm), likely due to the already very
low EC levels in the dry season in these small headwater catchments, indicative of fast, shallow flow paths and low weathering
rates all year round. DOC concentrations decreased from dry (median 2.9 mg/L) to wet seasons (median 2.2 mg/L) in the
agricultural sites, whereas there was high cross-site variability and no significant seasonal change in the rainforest sites.
Importantly, DOC concentrations varied in response to local flow conditions during the wet season. The highest three
concentrations (>5 mg/L) observed at sites C2, R5, and R6 coincided with the rising limb of a small high-flow event on March
25$^{th}$ (Figure S1), whereas all other sites were sampled under flow recession, resulting in depleted DOC sources and lower
concentrations.

The $\delta^{13}$C-DOC values were always lower than -27‰ and relatively stable between seasons and across land use categories,
although there was a notable difference (p=0.053) between rainforest sites (median -31.0‰) and agricultural sites (median -
28.9‰) in the wet season. BIX values significantly decreased from dry (median 0.86) to wet seasons (median 0.61), while
HIX values were generally low (< 4.1) and did not vary consistently with season. Neither BIX nor HIX values were
significantly different between land use categories.

We found that the age of exported DOC was not consistent among streams draining the same land use category, with no
significant difference between land use categories during the dry season (p=0.398), wet season (p=0.073), or both seasons
combined (p=0.589). However, the response of individual sites to high flow conditions diverged between the two land use
categories (Figure 3a). In rainforest catchments, high flows tended to mobilise younger DOC compared to baseflow conditions
(p=0.062); this pattern was observed at five out of six sites. In contrast, in agricultural catchments, high flows tended to
mobilise DOC of similar age or older compared to dry season flows (p=0.129); a pattern that was observed at eight of the ten
sites with repeated measurements. The fraction of rainforest cover explained 29% of the variability in these seasonal changes,
with sites with more forest cover having a more positive change in DOC age between seasons, i.e. younger DOC in the wet
season (Figure 3b).





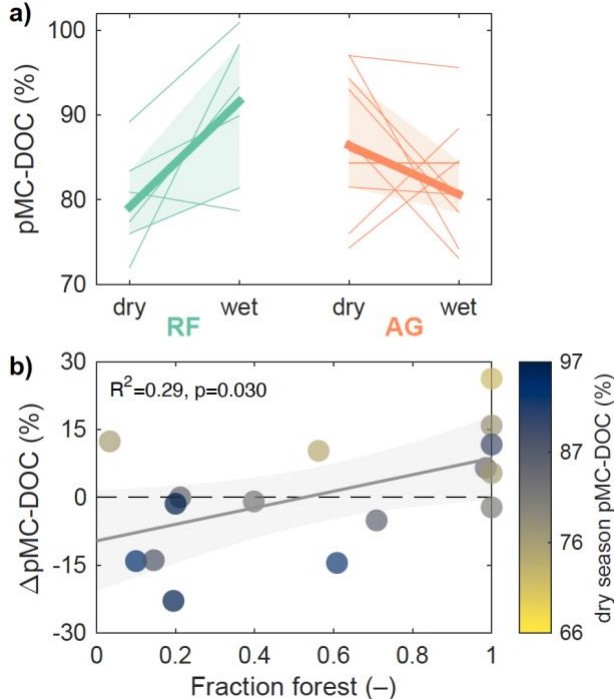

**Figure 3. Seasonal changes in percent modern C (pMC) in dissolved organic carbon (DOC). a) Changes in pMC-DOC for the six rainforest (RF) sites and ten agricultural (AG) sites. The thick lines and shaded areas represent the median and interquartile range across all sites. b) Relationship between seasonal change in pMC from dry to wet seasons (ΔpMC-DOC; positive = younger DOC in the wet season) and fraction of forest cover. All six rainforest sites plot to the right around 1. None of the temporal or cross-category changes in the upper plot are significant at a 95% confidence level, although the seasonal change in rainforest sites has a low p-value (p=0.062), as does the difference between the two land use categories during the wet season (p=0.073; Figure 2).**

The drivers of variation and relationships with DOC age differed between the dry and wet seasons (Figure 4). In the dry season, catchment area, water temperature, DOC concentrations, $\delta^{13}$C-DOC, and BIX were positively correlated with pMC-DOC, indicating that higher values of these variables were associated with younger DOC, while land use * slope and HIX were negatively correlated with pMC-DOC. In the wet season, the most significant drivers were land use * slope, HIX, and DOC concentrations, all positively correlated with pMC-DOC, while catchment area was negatively correlated with pMC-DOC. While the two land use categories did not explain significant differences in DOC age (Figure 2), the GAM results indicate that the interaction between slope and land use was a strong driver of pMC-DOC in both seasons. Models that treated land use and slope as separate predictors resulted in higher AIC values (-407 and -350 for the dry and wet seasons, respectively) compared to GAMs that included the land use * slope interaction (AICs of -427 and -360 for the dry and wet seasons, respectively), suggesting that the interaction term improved model performance.



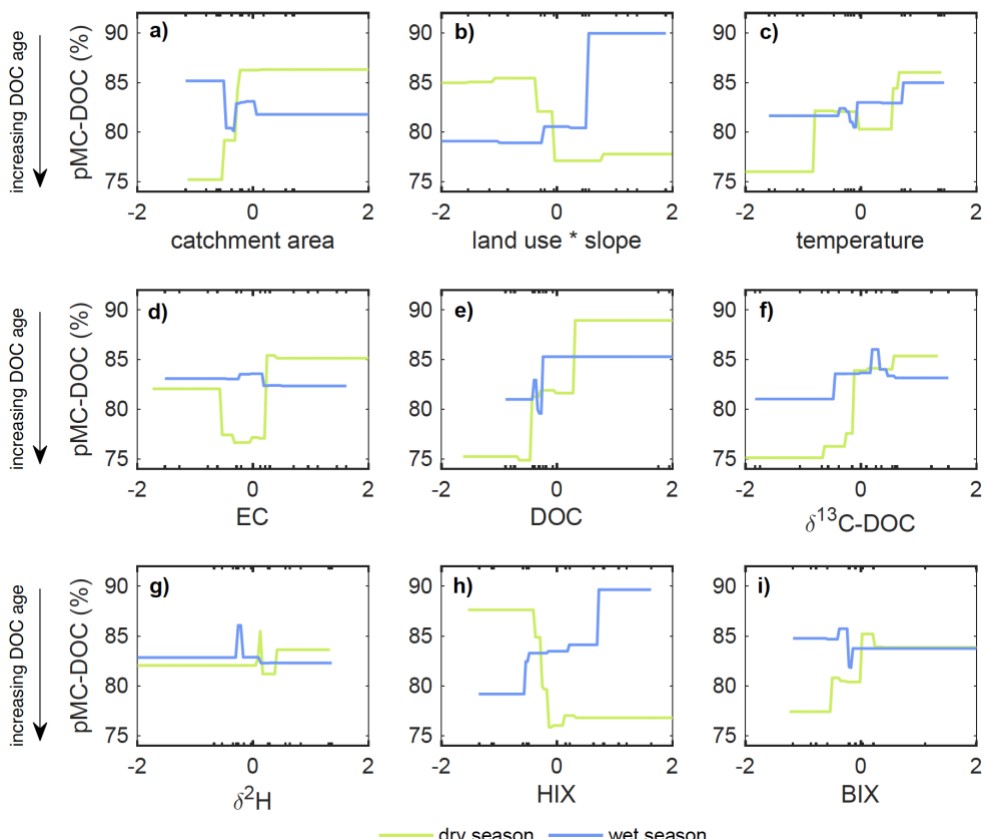

**Figure 4. Partial dependence plots of the marginal effects of each predictor on $^{14}$C-DOC based on the best GAMs developed for the dry and wet seasons. All predictors were standardised prior to analysis, so that negative values on the x-axes indicate below-mean values, while positive values indicate above-mean values. Note: higher pMC-DOC corresponds to lower DOC age.**

255

The DIC samples collected at three sites were modern or close to modern, with mean values of 97.1 pMC (R1), 98.5 pMC (C4) and 100.4 pMC (P1). Values slightly increased (i.e. DIC became younger) in the wet season at R1 and P1, whereas they decreased at C4. DIC was always younger than DOC, and the DIC and DOC ages varied consistently across seasons (Figure 5). However, these seasonal changes occurred in opposite directions at the rainforest site (R1), with younger DIC and DOC in the wet season, compared to the agricultural site (C4), with older DIC and DOC in the wet season. Unfortunately, one $^{14}$C-DOC sample from the third site (P1) was lost during the analysis, so we have no DOC age data to compare the wet season $^{14}$C-DIC sample to. Using our simple endmember analysis, we estimated that between 13 and 65% (mean 35%) of DIC might have originated from in-stream mineralisation of DOC, with the rest likely originating from soil $CO_2$.





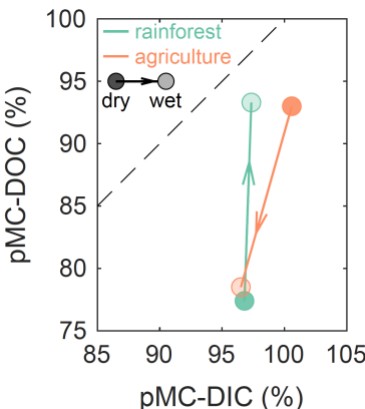

**Figure 5. Concomitant changes in percent modern C (pMC) in DIC and DOC for two streams representative of the two broad land use categories (R1; rainforest and C4; agricultural). The dashed line represents the 1:1 line, along which DIC and DOC would have equivalent ages.**

## 4    Discussion

### 4.1    Controls on DOC age in rainforest versus agricultural streams

We observed high variability in DOC age across sites, with no consistent differences between rainforest and agricultural sites; a result that allows us to reject our first hypothesis that older DOC is exported after land conversion to agriculture. A range of factors may explain this cross-site variability. First, our two soil cores had very distinct age profiles (Table S3), with SOC at > 20 cm depth being either modern or millennia old. Such discrepancy may be linked to geomorphic controls: the R6 core, which contained modern SOC at depth, was taken from a low point near the stream, where organic matter likely accumulates more rapidly. In contrast, the R3 core, which contained old SOC at depth, was taken from a gentle slope. Such spatial variability in SOC age is not uncommon in Australian landscapes (Hobley et al., 2017; Bowman et al., 2004) and might have contributed to some of the differences in DOC age observed across sites. Given its location on a gentle slope, however, we believe the R3 core may be more representative of broader landscape conditions where DOC is produced and exported.

Hydrological flow paths also appeared to play an important role. The negative effect of DOC concentration on DOC age (years BP) in both seasons (Figure 4) suggests that some streams may be fed by slightly deeper subsurface contributions, with older and more processed, lower concentration DOC, whereas others were fed by shallower flow paths, which tend to contain younger and more concentrated DOC (Barnes et al., 2018; Campeau et al., 2019; Sanderman et al., 2009) as natural soil formation processes allow for a decreasing age with depth. Various factors can influence flow path depth, including seasonal changes in flow regime and catchment morphology. Steep catchments tend to have shallower flow paths and shorter residence times (Remondi et al., 2019; Tetzlaff et al., 2009), leading to the export of younger DOC. Steep forested catchments are also likely to generate high runoff containing freshly leached organic material from both throughfall (Van Stan and Stubbins, 2018; Behnke et al., 2023) and leaf litter (Janeau et al., 2014; Miranda and Avelar, 2022). In our study, slope was an important predictor of DOC age, particularly in the wet season and for rainforest sites, where steeper catchments tended to export younger





DOC ($R^2$=0.54; p=0.36). A similar influence of catchment morphology on DOC age has been observed in subtropical
mountainous catchments (Chen et al., 2023). Interestingly, DOC ages (years BP) in the disturbed catchments had a negative
(though not statistically significant) relationship with slope in both seasons, contrasting with the positive relationship observed
in rainforest catchments. This finding suggests that in agricultural catchments, the rate and magnitude of soil erosion, rather
than flow path depth, may be a key control on DOC ages. In these systems, steeper catchments may increase the mobilisation
of aged soil C, likely exposed by tillage and/or cattle trampling (Sickman et al., 2010; Drake et al., 2019).

The sources of DOC were relatively similar between rainforest and agricultural catchments, with $\delta^{13}$C-DOC compositions
indicating similarly depleted, likely C3 plant-derived biomass as the dominant source. This is despite the agricultural
catchments having available C4 pools (sugar cane and some pasture grass species), and suggests that most DOC in disturbed
catchments originated from older, pre-clearing soil C pools – a result similar to that of Moyer et al. (2013). Lower contributions
of C4-derived C sources than would be expected based on land use have been related to potentially large contributions from
riparian forests and to differences in the biolability of different DOC sources (Bouillon et al., 2012; Bouillon et al., 2007).
While we lack $\delta^{13}$C-SOC data to confirm the importance of older soil C sources to the stream DOC pool in disturbed
catchments, the negative relationship between $\delta^{13}$C-DOC and DOC age (years BP) in the dry season (Figure 4), and the
generally older wet-season DOC ages at these sites, both tend to support our conclusion (see following Section). The relatively
low HIX values across sites and seasons indicate that regardless of DOC age, highly biolabile compounds, which are typically
low molecular weight and less aromatic in nature (Berggren et al., 2009; Li et al., 2023), may be dominating the DOC pool.
This is not unexpected, as previous studies have shown that aged C can also be relatively enriched in energy-rich organic
matter at the molecular level and be highly biolabile and readily oxidised by microbes (Mann et al., 2015; Drake et al., 2019;
Spencer et al., 2015).

Despite differences across sites and no clear distinction between DOC ages in rainforest *versus* agricultural catchments, our
data revealed some unexpected findings. In contrast to most previous tropical studies, we found that DOC in small, headwater
streams of humid tropical regions can be centuries to millennia old (with a median of 82.5 pMC or 1,553 years BP across sites
and seasons), even in pristine rainforest areas. Recent studies have reported similar occurrences of old DOC at low flow in
small, relatively undisturbed catchments in temperate regions (Tittel et al., 2022; Rhyner et al., 2023). Our findings suggest
that this pattern may also apply to tropical regions, challenging the view of faster DOC turnover in the tropics (Spencer et al.,
2012; Mayorga et al., 2005; Müller et al., 2015; Marwick et al., 2015). One possible explanation for this result is the higher
elevations (hence lower temperatures) of our six pristine sites (mean elevations 345–948 masl; *Cfa* climate class), which may
have led to relatively slower decomposition rates and older SOC accumulating over time, compared to what would be expected
in warmer, lowland tropical areas. Another potential factor is the contribution of old groundwater-derived DOC to streams
(Mcdonough et al., 2020), a possibility we discuss further in Section 4.3.





## 4.2 Seasonal shifts in DOC age and pathways

Our results show that DOC age varied significantly between dry and wet seasons, with marked differences between rainforest catchments, where DOC tended to be younger at high flow, and agricultural catchments, where DOC was similar or older at high flow (Figure 6). These findings support our second hypothesis that DOC is younger at high flow for rainforest sites, but contradict it for agricultural sites. We interpret these differences between land use categories as reflecting fundamental differences in system dynamics. Rainforest sites were in an 'equilibrium' state, where DOC age was a direct function of flow path depth and reflected the hydrological shift between dry and wet seasons. This pattern is well documented in temperate and northern high-latitude catchments, where during high flow conditions the shallower riparian or organic-rich surface soil and litter layers become connected to streams and mobilise younger DOC (e.g. Tittel et al., 2015; Barnes et al., 2018; Evans et al., 2022; Campeau et al., 2019; Aiken et al., 2014). Additionally, the younger DOC observed in the wet season may also be attributed to increased leaching of fresh leaf litter (abundant in the rainforest catchments) through surface flow pathways, as reported in similar steep tropical rainforest regions (Miranda and Avelar, 2022). Leaf litter is a source of young, unprocessed, low molecular weight, biolabile C (e.g. Hudson et al., 2018; Harfmann et al., 2019), and the low HIX and BIX values observed at our rainforest sites during the wet season are a strong indication that leaf litter leaching contributed to DOC in these streams.

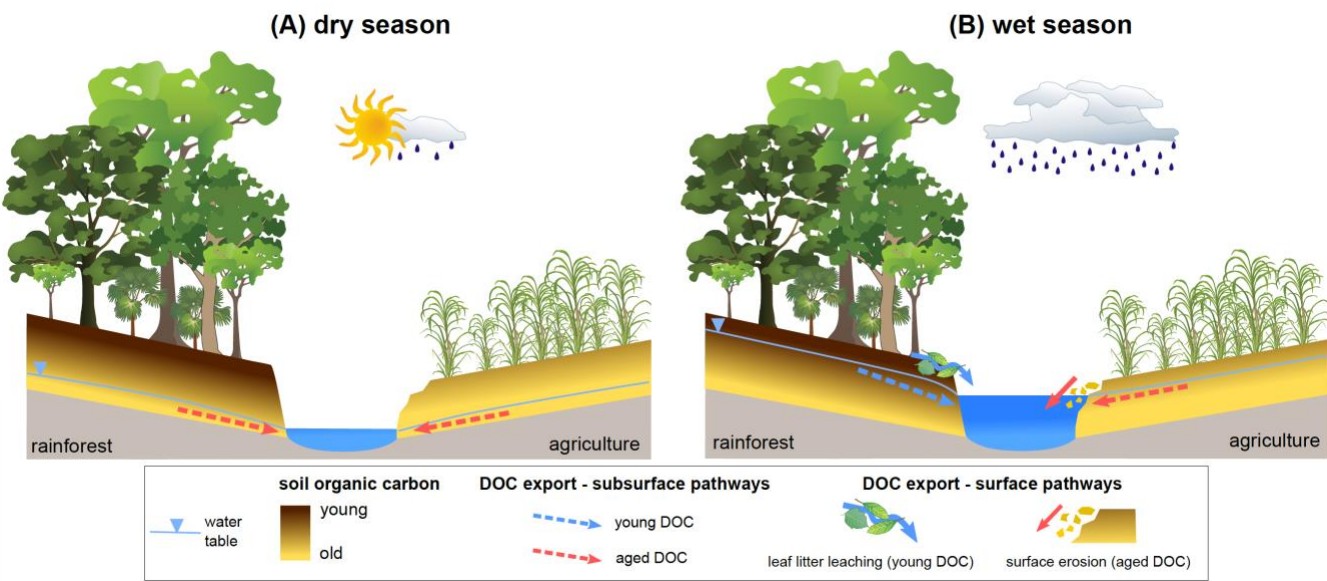

**Figure 6. Conceptual diagram showing the shift in DOC sources between the (A) dry and (B) wet seasons in rainforest *versus* agricultural catchments.**

In contrast, agricultural sites operated under unsteady-state conditions, where old DOC was exported regardless of flow conditions. This likely results from anthropogenic disturbance removing the bulk of modern C from organic-rich surface soil and litter layers (Drake et al., 2019; Moore et al., 2013; Lee et al., 2021). At some agricultural sites, higher flow conditions even led to the export of older DOC, a result that aligns with Moore et al. (2013) who observed the export of older DOC during




the wet season from two disturbed peat catchments in Borneo. We propose that at these sites, processes such as tillage, cattle trampling and bank erosion may have exposed deeper, older C at the surface, increasing the leaching and transport of older DOC via surface rather than subsurface pathways during the wet season (Figure 6). While the role of surface erosion on DOC

export is unclear, studies have reported enhanced DOC mobilisation with increased erosion (Van Gaelen et al., 2014; Chen et al., 2023). Alternatively, older DOC at higher flow could also originate from $^{14}$C-dead organic chemicals transported via runoff (Sickman et al., 2010), such as synthetic fertilisers commonly used in sugarcane cultivation – however, we suggest this explanation is likely not a major control on DOC age across our sites, as it would not apply to pasture sites where fertilisers are not used.

In the wet season, the two land use categories had relatively different DOC sources (p=0.053), with more depleted values (median ~-31‰) in rainforest sites and slightly more enriched values (median ~-29‰) in agricultural sites. These differences may be related to increased contributions of young, C3 plant-derived organic C in the rainforest sites compared to older SOC in agricultural sites, as the latter is typically more $^{13}$C enriched than C3 plant-derived biomass (Moyer et al., 2013; Drake et al., 2019). These apparent differences in DOC sources were not reflected in the BIX and HIX indices, where seasonal variations

were more pronounced than the differences between land use categories (Figure 2). The decrease in BIX values from dry to wet seasons suggests a shift from a mixture of aquatic/microbial and terrestrial DOM sources to mostly terrestrial sources (Huguet et al., 2009; Fellman et al., 2010). This shift is consistent with increasing hydrological connectivity between soil C pools and streams, and in some cases, with higher surface erosion and runoff. The low HIX values across all sites indicate low humification, which is relatively common in tropical streams and rivers (Ji et al., 2024; Iles et al., 2022; Lambert et al., 2016).

This suggests that even older SOC pools underwent limited humification before being mobilised into streams as DOC. Additionally, at sites with gentler slopes, slower streamflow and greater evaporation, older C might have been photodegraded over time, transforming aromatic-rich organic matter to lower molecular weight, more biolabile moeities (Stubbins et al., 2010). Alternatively, the dominance of low molecular weight DOM could suggest potential connection to regional groundwater systems at certain sites, as groundwater has been shown to contain old C with low molecular weight (Mcdonough et al., 2020).

This explanation is explored further in the following section.

### 4.3    Decoupling between DIC and DOC cycling

Unlike DOC, DIC was mostly modern (96.5 < pMC-DIC < 101.2) and always younger than DOC, consistent with our third hypothesis that DIC is younger than DOC in the study region. At the three sampled sites (R1, P1 and C4), DIC was primarily sourced from young, soil-derived $CO_2$ inputs to streams (35–87%) rather than from internal respiration of older DOC. These

results suggest a decoupling between DIC and DOC cycling, regardless of land use. Consistent with our work, previous studies have also observed a disconnect between the sources and controls of DIC and DOC in streams, as reported in both boreal and tropical areas (Campeau et al., 2019; Moyer et al., 2013). Younger $CO_2$/DIC has been associated with the lateral export of fresh soil $CO_2$ (Campeau et al., 2019) or to direct inputs of atmospheric $CO_2$ (Moyer et al., 2013). The dominance of soil-derived $CO_2$ inputs to the stream DIC pool is well-documented in Australian tropical streams (Solano et al., 2024; Duvert et



al., 2020) and, more generally, in low-order streams (Hotchkiss et al., 2015; Battin et al., 2023). However, in-stream DOC respiration can also be a substantial source of stream DIC (e.g. Solano et al., 2023), and it is interesting to note that even in these relatively steep headwater streams characterised by short residence times, stream DOC mineralisation contributed a non-negligible fraction of the stream DIC pool. This may have been facilitated by potentially high DOM biolability across sites, as indicated by the low HIX values, and relatively high BIX values, particularly in the dry season.

Groundwater is increasingly recognised as a potentially significant source of old, biolabile DOC to surface systems (Mcdonough et al., 2022; Mcdonough et al., 2020). There is a possibility that at our sites, some old DOC could have entered streams via groundwater rather than solely through the mobilisation of old soil organic C. However, the DIC data suggest this is unlikely, as DIC remained modern or near-modern in both the wet and dry seasons. Any groundwater contributions to streams likely originated from shallow flow paths composed of recently recharged waters, as further corroborated by the low

EC values across all sites and seasons and the significant seasonal changes in water stable isotopes, which suggest minimal regional groundwater influence and short residence times in the subsurface. Our DIC dataset includes only three of the 18 sites, so additional DIC age data would be necessary to definitively rule out groundwater as a source of old DOC. This would be particularly relevant in larger, flatter catchments where older, regional groundwater contributions may be more prevalent.

## 5    Conclusions

Our study shows that DOC exported from tropical headwater streams can be old, particularly during drier periods, challenging the common view of fast SOC turnover in tropical systems. We also highlight the influence of seasonal variations on DOC age, with flow path depth and leaf litter layer leaching through wet-season runoff being the dominant factors in pristine rainforest catchments, with DOC becoming younger at high flow. In contrast, in agricultural catchments, DOC age was primarily influenced by the loss of modern C due to land conversion and contemporary soil erosion, particularly at high flow.

Lastly, our findings suggest that the stream DIC and DOC pools are largely decoupled in these systems, with DIC being younger than DOC and in-stream respiration of DOC representing a minor fraction of the DIC pool.

While our findings provide valuable new insights, there is still much to explore. In particular, our wet season sampling occurred three weeks after the last major rainfall event (Figure S1) and during flow recession conditions, when soil DOC sources may have already been depleted. Clearly, more frequent sampling will be required to provide a clearer picture of the temporal

dynamics of DOC age at different stages of the wet season.

Climate change is expected to intensify hydrological extremes, potentially leading to increased erosion and depletion of soil C stocks in agricultural areas. Understanding the extent to which old soil C is reintroduced into the modern C cycle under these changing conditions will be crucial for predicting long-term C fluxes and ecosystem responses, particularly in disturbed tropical landscapes.



## 6    Data availability

The dataset used in this study is available at https://www.hydroshare.org/resource/c5e20d5ffe5441e2bf54ba0561fb4dd4/

## 7    Code availability

The code used in this study is available upon request from the corresponding author.

## 8    Acknowledgments

We acknowledge financial support from an AINSE 2019 Early Career Researcher Grant and a CDU Covid Supplementary Funding Project, both awarded to CD. Our thanks go to Peter Freeman, Vladimir Levchenko (ANSTO), Matheus Carvalho (Southern Cross University) and Stewart Fallon (Australian National University) for assistance with laboratory work, to Rainy Comley and Michael Bird (James Cook University) for support with field logistics, and to Paula Martínez for fieldwork assistance. CD is supported by a fellowship from the Australian Research Council (DE220100852).

## 9    Author contributions

CD, DEB, DIC and RGMS designed the study. CD conducted fieldwork with assistance from NCM for site selection. VS performed the geospatial analyses. LMD performed the DOM composition analyses. CD analysed and interpreted the data with inputs from LMD, VS and FUC. CD wrote the initial draft which all authors reviewed and edited.

## 10    Competing interests

The authors declare that they have no conflict of interest.

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
