# Peer review of "Tropical wet season runoff mobilises younger carbon in rainforest streams but older carbon in agricultural streams"

_EGUsphere, 2025_

## Referee Comment (RC2)

Duvert et al

Duvert et al. surveyed agricultural and forested headwater watersheds in tropical Australia to address the question: 'How does land use change and flow conditions affect the age of DOC exported from these watersheds? Hypotheses to be tested are:
- H1- land use conversion to agriculture leads to the export of older DOC because erosion has depleted modern soil OC
- H2- younger DOC is exported during high flow conditions compared to baseflow because high flows mobilize younger OC from shallower soil layers.
- H3- stream DIC is younger than DOC because DIC is derived from root respiration and decomposition of younger, more labile DOC (i.e., DIC and DOC ages are uncoupled).

As the authors note, the tropics are understudied with respect to carbon loss in general, and in response to agricultural activities in particular. Thus, this is a valuable question to explore.

The cost of analysis means that sample sizes are understandably small for studies based on 14C data. The authors have supplemented their 14C-DOC results with other measures of DOC quality, but in the end, they are still constrained by a small sample size (N = 18 and 16 for their dry and wet season datasets, respectively - or 6 and 12 forest and agricultural sites).

The key set of results are presented in Fig. 2, with statistical results of comparisons between land use types for a given season, and seasonal effects within each land use category. There is a fair amount of variability in the results, thus the authors used generalised additive models (GAMs) informed by 9 driver variables to identify the sources of this variability. This choice is problematic, as GAMs and other ML approaches are built for use with large data sets. The effective degrees of freedom for non-linear relationships GAMs tend to be high. A common rule of thumb for various (less complicated) regressions is 10 observations per parameter (Harrell 2015), suggesting that an appropriate sample size for 9 covariates would be well into the hundreds and beyond. Thus, I am skeptical of the robustness of results presented in Fig. 4 and wondered if confidence intervals were excluded from these plots because they were large. The authors should give serious consideration to removing this analysis from the manuscript.

It took me a while to sort through the results and discussion, perhaps not surprising given the fact that I do not work with isotope data on a regular basis combined with the consideration of 9 different response variables across two land uses x two seasons, and the reality that interpretation of isotopic data can be tricky because multiple sources and processes may shape these values. Unfortunately, this complexity was enhanced by occasional contradictory or slightly misleading statements and introduction of new results in the Discussion (see line-specific comments for examples). Figure 6 is a very nice way to summarize the authors' hypothesis for the differences in DOC age shown in Fig. 2g. It makes sense, but there are some missing or ambiguous pieces of evidence in its support- mostly for the agricultural side of the figure, and at times it felt as if the authors were working hard to make their data fit into this model alone. The most obvious evidence gap is the assumption that soil C in agricultural areas is old at all depths. Soil erosion is a reasonable mechanism for moving soil C to the stream in agricultural streams, but evidence that this is happening is lacking (see comment for line 290). As the Duvert et al. acknowledge, their rainforest sites are steeper and they also tend to occur at higher elevations than agricultural

sites. Agricultural areas are often former grasslands, which have distinct structure and carbon storage compared to forests. This leads to the question: is it agricultural disturbance that is creating the differences reported here, or is it something to do with these differences in physical attributes and/or a legacy of past land cover?

Regardless of mechanism, the age differences shown in Fig. 2g and the finding that DOC in these sites is unexpectedly old are intriguing results. Because I am a non-regular user of 14C data, I was interested in having some context for the old DOC in these streams. This led me to Table 1 in Shi et al. (2020). Put another way, is there any contextual data available for considering DOC ages presented here?

Line 46- It seems like the recent paper by Dean et al. (2025) should be acknowledged/incorporated into this paragraph.

Line 71- 'The humid tropics of Australia is a mountainous region…

Line 226-228- This sentence is confusing- it starts by stating that there were no differences for streams draining the same land use category (i.e., within land use type), but then it goes on 'with no differences between land use categories…'.

Also, here you report that there is no significant difference between land uses during the wet season (p = 0.073). In other locations in the paper, p values slightly greater than 0.05 are noted as indicating differences (e.g., lines 222, 230, 231, and in a slightly different fashion, line 291). Given the small sample size in this study, I think some leniency on p values would be appropriate. Consider adding some text to the methods such as 'p < 0.10 were viewed as being indicative of differences, given the small sample size.' But whether or not this is done, there needs to be some consistency in how these statistical results are reported.

Line 230-231- The statement that "in agricultural catchments, high flows tended to mobilise DOC of similar age or older compared to dry season flows (p=0.129)" feels a bit misleading. Fig 2g shows samples with wet season DOC ages that are both older and younger than some of the DOC collected during the dry season. By itself, this figure indicates that the age of DOC in agricultural streams did not differ between seasons. And with respect to the comment about p values above, this one seems a bit too high to hint at a meaningful environmental difference.

Line 231-232- I am also confused by '8 of 10 sites with repeated measures' here. Weren't all 12 agricultural streams sampled twice, per lines 119-120? Can you clarify?

Fig 4 (if it is retained)- Please explain the barely visible dashes on the upper and lower horizontal axes. Typically, these dashes indicate the number of samples at each predictor value, and I first assumed that 1 set of dashes (e.g., on the top of each plot) represented predictor values during one season/model, and the second set (along the bottom) corresponded to the other season/model. However, the distribution of predictor values on top and bottom look the same. If these dashes are to be included, then they need some explanation. These dashes can be useful,

particularly given that the 'curves' do not have any confidence intervals. I worry that the confidence intervals were omitted because the small sample size led to wide confidence intervals, indicative of a weak model.

Line 258- unclear what is meant by 'DIC and DOC ages varied consistently across seasons' given that changes in DOC ages were in opposite direction between seasons.

Line 280- Line 243 states that pMC-DOC was positively correlated with DOC concentration in the dry season, per Fig. 4. But there is no similar statement about a negative effect of DOC concentration on DOC age during the wet season though, which is not surprising given that panel 4e suggests that this relationship is fairly weak. Thus, this statement is new news. Table S2 (which is mentioned only in the Methods section) does show a significant relationship between DOC concentration and age during the wet season, but does not indicate the direction of this relationship. I was curious, so I plotted the data with a loess smoother (+/1 1SE):

[Figure]

I understand that meaningful relationships identified in a GAM are not always apparent in a univariate plot, but it is hard to relate the plot above to the argument that increasing DOC concentrations are associated with decreasing DOC age during the dry season. The argument that older DOC reflects deeper flow paths is also a bit at odds with the age distribution of carbon in core R6 (the one that was considered not to be representative).

Lines 287-289- This is another new result and looks to be from a linear regression of slope vs. DOC age for wet season rainforest sites (for full transparency, I ran this linear regression and got the same R2 as reported here). The R2 values is at odds with the p value, but fortunately, the former is more meaningful than the latter, given N = 6.

Lines 290-294- This is another new result. Given the indication of no relationship between slope and DOC age from Table S2, I generated more plots- with agricultural sites pooled and not pooled for the 2 seasons (this time, confidence intervals were generated using a linear model). Again, I recognize that relationships in partial dependency plots are not necessarily apparent in

univariate plots, and that land*slope was used in the GAM. However, this sentence is talking specifically about slope, as is the case for results reported on line 289.

In both cases, these relationships are basically flat and do not support the statement made here that there may be a negative relationship between slope and DOC age.

[Figure]

Lined 306-308- This sentence about aged C being highly biolabile surprised me. In part, it was unexpected because of conventional wisdom that more labile molecules are lost first because they are easier to mineralize, but also because this view is the basis for your third hypothesis (DIC is younger than DOC due to external inputs of young soil CO2 and to younger, more biolabile organic matter being preferentially mineralized). Similarly, leaf litter leachate is described as young and biolabile (lines 331-332).

Line 338- Yes, relatively old DOC was exported during both wet and dry conditions in agricultural streams. But the wording is potentially confusing in the context of Fig. 3a and line 344 that suggest that flow conditions do affect DOC age. Can this be clarified to minimize possible confusion?

Harrell, F.E. 2015. Regression modeling strategies. Springer Nature Publishing.

---

## Author Comment (AC2)

**Referee #2**

We thank referee #2 for their thorough evaluation of this work. Our responses to their comments are in blue.

Duvert et al. surveyed agricultural and forested headwater watersheds in tropical Australia to address the question: 'How does land use change and flow conditions affect the age of DOC exported from these watersheds? Hypotheses to be tested are:

- H1- land use conversion to agriculture leads to the export of older DOC because erosion has depleted modern soil OC
- H2- younger DOC is exported during high flow conditions compared to baseflow because high flows mobilize younger OC from shallower soil layers.
- H3- stream DIC is younger than DOC because DIC is derived from root respiration and decomposition of younger, more labile DOC (i.e., DIC and DOC ages are uncoupled).

As the authors note, the tropics are understudied with respect to carbon loss in general, and in response to agricultural activities in particular. Thus, this is a valuable question to explore.

The cost of analysis means that sample sizes are understandably small for studies based on 14C data. The authors have supplemented their 14C-DOC results with other measures of DOC quality, but in the end, they are still constrained by a small sample size (N = 18 and 16 for their dry and wet season datasets, respectively - or 6 and 12 forest and agricultural sites).

We agree that the sample sizes in our study are limited, which makes our interpretations less robust. But as the referee notes, we believe there is still value in presenting the observed trends in our dataset.

The key set of results are presented in Fig. 2, with statistical results of comparisons between land use types for a given season, and seasonal effects within each land use category. There is a fair amount of variability in the results, thus the authors used generalised additive models (GAMs) informed by 9 driver variables to identify the sources of this variability. This choice is problematic, as GAMs and other ML approaches are built for use with large data sets. The effective degrees of freedom for non-linear relationships GAMs tend to be high. A common rule of thumb for various (less complicated) regressions is 10 observations per parameter (Harrell 2015), suggesting that an appropriate sample size for 9 covariates would be well into the hundreds and beyond. Thus, I am skeptical of the robustness of results presented in Fig. 4 and wondered if confidence intervals were excluded from these plots because they were large. The authors should give serious consideration to removing this analysis from the manuscript.

These are very important points raised by the referee. Our choice of nine predictors may indeed have been too high given the limited number of observations. We have rerun the GAMs, this time removing more collinear predictors with Pearson coefficients > 0.5, resulting in models with five predictors. We have also added confidence intervals (interquartile ranges) to the updated model outputs, calculated from 100 bootstrap iterations of the GAM predictions:

We believe the updated GAM results are more robust than the earlier version. However, if the referee and editor consider that the analysis still yields results that are too uncertain, we would consider removing this aspect of the ms.

It took me a while to sort through the results and discussion, perhaps not surprising given the fact that I do not work with isotope data on a regular basis combined with the consideration of 9 different response variables across two land uses x two seasons, and the reality that interpretation of isotopic data can be tricky because multiple sources and processes may shape these values. Unfortunately, this complexity was enhanced by occasional contradictory or slightly misleading statements and introduction of new results in the Discussion (see line-specific comments for examples). Figure 6 is a very nice way to summarize the authors' hypothesis for the differences in DOC age shown in Fig. 2g. It makes sense, but there are some missing or ambiguous pieces of evidence in its supportmostly for the agricultural side of the figure, and at times it felt as if the authors were working hard to make their data fit into this model alone. The most obvious evidence gap is the assumption that soil C in agricultural areas is old at all depths. Soil erosion is a reasonable mechanism for moving soil C to the stream in agricultural streams, but evidence that this is happening is lacking (see comment for line 290). As the Duvert et al. acknowledge, their rainforest sites are steeper and they also tend to occur at higher elevations than agricultural sites. Agricultural areas are often former grasslands, which have distinct structure and carbon storage compared to forests. This leads to the question: is it agricultural disturbance that is creating the differences reported here, or is it something to do with these differences in physical attributes and/or a legacy of past land cover?

First, we wish to clarify that all the agricultural sites in our study were historically covered by rainforest – either lowland rainforest in the flatter areas now used for sugarcane crops, or upland rainforest in the steeper areas now used for pasture. There is ample evidence that extensive rainforest clearing occurred in these regions from the late 1800s to support the sugarcane and grazing industries (Birtles, 1982; Kemp et al., 2007; Vanclay, 1996). While some lowland open forest and pockets of grassland occurred in the area, rainforest was by

far the most common and widespread vegetation type in the Innisfail region, where our agricultural sites are located (Kemp et al., 2007).

We agree that our study provides only indirect evidence that soil carbon in agricultural areas is old. However, there is a long history of erosion in these catchments, which has been well documented as a major environmental issue for the Great Barrier Reef region (e.g. Kroon et al., 2016; McKergow et al., 2005). Given this persistent erosion issue, it is reasonable to expect the mobilisation of older soil C from agricultural sites. That said, we cannot fully rule out the possibility that the observed differences in DOC age also reflect contrasts in soil properties across sites. We will add this caveat in the revised manuscript.

Regardless of mechanism, the age differences shown in Fig. 2g and the finding that DOC in these sites is unexpectedly old are intriguing results. Because I am a non-regular user of 14C data, I was interested in having some context for the old DOC in these streams. This led me to Table 1 in Shi et al. (2020). Put another way, is there any contextual data available for considering DOC ages presented here?

Thanks for the suggestion to put our results in the context of SOC age studies. We will add a short comparison with values from Shi et al. (2020), as well as studies conducted in the tropics (Drake et al., 2019) and in Australia (Bowman et al., 2004; Hobley et al., 2017). In short, aged SOC is not uncommon at relatively shallow depths, with Hobley et al. (2017) showing that SOC > 1,000 years BP dominates at depths > 30-40 cm under native vegetation and at depths > 10-15 cm in cropland soils.

**Specific comments**

Line 46- It seems like the recent paper by Dean et al. (2025) should be acknowledged/incorporated into this paragraph.

Yes, we'll add this ref.

Line 71- 'The humid tropics of Australia is a mountainous region...

**OK**

Line 226-228- This sentence is confusing- it starts by stating that there were no differences for streams draining the same land use category (i.e., within land use type), but then it goes on 'with no differences between land use categories...'.

**This will be rephrased.**

Also, here you report that there is no significant difference between land uses during the wet season (p = 0.073). In other locations in the paper, p values slightly greater than 0.05 are noted as indicating differences (e.g., lines 222, 230, 231, and in a slightly different fashion, line 291). Given the small sample size in this study, I think some leniency on p values would be appropriate. Consider adding some text to the methods such as 'p

I understand that meaningful relationships identified in a GAM are not always apparent in a univariate plot, but it is hard to relate the plot above to the argument that increasing DOC concentrations are associated with decreasing DOC age during the dry season. The argument that older DOC reflects deeper flow paths is also a bit at odds with the age distribution of carbon in core R6 (the one that was considered not to be representative).

The relationships between DOC and pMC-DOC are actually detailed in the Results (L242-243 for the dry season and L245-246 for the wet season). We acknowledge that the dry-season relationship is not clear cut given the influence of an outlier (DOC  $\sim$  5 mg/L, low pMC-DOC). However, despite this trend being not visually apparent, the GAM does suggest a strong negative relationship between DOC concentration and DOC age during the dry season – see updated results in our plot a few pages above. We will clarify this point in the text and acknowledge that this trend is weak in the raw data (with a reference to Table S2) and not consistent across all sites.

Lines 287-289- This is another new result and looks to be from a linear regression of slope vs. DOC age for wet season rainforest sites (for full transparency, I ran this linear regression and got the same R2 as reported here). The R2 values is at odds with the p value, but fortunately, the former is more meaningful than the latter, given N = 6.

Sorry for introducing this result in the Discussion. We will move this to the Results.

Lines 290-294- This is another new result. Given the indication of no relationship between slope and DOC age from Table S2, I generated more plots- with agricultural sites pooled and not pooled for the 2 seasons (this time, confidence intervals were generated using a linear model). Again, I recognize that relationships in partial dependency plots are not necessarily apparent in univariate plots, and that land\*slope was used in the GAM. However, this sentence is talking specifically about slope, as is the case for results reported on line 289.

In both cases, these relationships are basically flat and do not support the statement made here that there may be a negative relationship between slope and DOC age.

This is a fair assessment. We will tone down the statement L290–294 and clarify that no clear relationship between slope and DOC age is evident in the raw data (Table S2). We will also note that the apparent trend in the GAM likely reflects interactions with land use rather than the sole effect of slope.

Lined 306-308- This sentence about aged C being highly biolabile surprised me. In part, it was unexpected because of conventional wisdom that more labile molecules are lost first because they are easier to mineralize, but also because this view is the basis for your third hypothesis (DIC is younger than DOC due to external inputs of young soil CO2 and to younger, more biolabile organic matter being preferentially mineralized). Similarly, leaf litter leachate is described as young and biolabile (lines 331-332).

We agree that the statement could sound counterintuitive – but several studies (e.g. Mann et al., 2015; Drake et al., 2019; Spencer et al., 2015) have demonstrated that aged DOC can still contain labile and energy-rich compounds. To avoid any confusion, we will slightly reword the third hypothesis: "Here we hypothesised that DIC is younger than DOC, due to external inputs of recent soil CO2 and the preferential mineralisation of biolabile organic matter (while recognising that DOC age and biolability do not always covary)."

Line 338- Yes, relatively old DOC was exported during both wet and dry conditions in agricultural streams. But the wording is potentially confusing in the context of Fig. 3a and line 344 that suggest that flow conditions do affect DOC age. Can this be clarified to minimize possible confusion?

We agree that the original phrasing could be interpreted as implying that flow conditions had no influence on DOC age. We will revise the sentence to clarify this, e.g.: "In contrast,

agricultural sites operated under unsteady-state conditions, where old DOC was exported during both wet and dry periods, although flow conditions still influenced DOC age."

Harrell, F.E. 2015. Regression modeling strategies. Springer Nature Publishing.

**References**

- Birtles, T.G. 1982. Trees to Burn: Settlement in the Atherton-Evelyn Rainforest 1880-1900. North Australia Research Bulletin 8, 31-86.
- Bowman, D.M.J.S., Cook, G.D. and Zoppi, U. 2004. Holocene boundary dynamics of a northern Australian monsoon rainforest patch inferred from isotopic analysis of carbon, (14C and δ13C) and nitrogen (δ15N) in soil organic matter. Austral Ecology 29(6), 605-612.
- Drake, T.W., Van Oost, K., Barthel, M., Bauters, M., Hoyt, A.M., Podgorski, D.C., Six, J., Boeckx, P., Trumbore, S.E., Cizungu Ntaboba, L. and Spencer, R.G.M. 2019. Mobilization of aged and biolabile soil carbon by tropical deforestation. Nature Geoscience 12(7), 541-546.
- Hobley, E., Baldock, J., Hua, Q. and Wilson, B. 2017. Land-use contrasts reveal instability of subsoil organic carbon. Global Change Biology 23(2), 955-965.
- Kemp, J.E., Lovatt, R.J., Bahr, J.C., Kahler, C.P. and Appelman, C.N. 2007. Pre-clearing vegetation of the coastal lowlands of the Wet Tropics Bioregion, North Queensland. Cunninghamia 10(2), 285-329.
- Kroon, F.J., Thorburn, P., Schaffelke, B. and Whitten, S. 2016. Towards protecting the Great Barrier Reef from land-based pollution. Global Change Biology 22(6), 1985-2002.
- McKergow, L.A., Prosser, I.P., Hughes, A.O. and Brodie, J. 2005. Sources of sediment to the Great Barrier Reef World Heritage Area. Marine Pollution Bulletin 51(1), 200-211.
- Shi, Z., Allison, S.D., He, Y., Levine, P.A., Hoyt, A.M., Beem-Miller, J., Zhu, Q., Wieder, W.R., Trumbore, S. and Randerson, J.T. 2020. The age distribution of global soil carbon inferred from radiocarbon measurements. Nature Geoscience 13(8), 555-559.
- Vanclay, J.K. 1996. Lessons from the Queensland Rainforests. Journal of Sustainable Forestry 3(2-3), 1-27.

---

## Author Comment (AC3)

**Referee #1**

We thank the referee for their evaluation of our work. Our responses to their comments are provided in blue below.

In this manuscript, Duvert et al investigate the age of dissolved organic carbon (DOC) mobilised by streams in rainforest and agricultural dominated catchments during wet and dry seasons. Understanding how, and if, old OC is mobilised from landscapes with different land use/land cover is relevant for projections on global C budgets. The current changes in rainfall regimes, specially in the tropics, have also strong implications for C mobilisation and export. Yet, the number of studies addressing both spatial and temporal variations are limited. Considering this, the work of Duvert et al is timing. I found the manuscript well-organised and scientifically sound.

Key strengths of the manuscript: a) the hypothesis and how the authors address each one throughout the manuscript; b) the experimental design (study area and parameters measured); c) the number of samples for radiocarbon analysis as it is above the average found in for similar studies; d) the use of H and O stable isotopes in water to link the hydrological and C cycle, and e) data analysis, presentation and interpretation.

**Thanks for this very positive assessment.**

The main limitation of this work is the calculation of the fraction of mineralised DOC that contributes to DIC. The results indicate that that between 13-65% of the DIC originates from in-stream mineralisation of DOC (line 262). Given the low number of samples analysed for DIC (only three samples per season), the uncertainty around those estimates is rather high. Moreover, the mixed-model approach to calculate the fraction of DOC being mineralised appears not robust enough. The authors partially acknowledge this limitation (lines 190-195). Although estimating these contributions is interesting, it appears that it's beyond the scope of the manuscript. Hypothesis #3 refers to a comparison between the age of DOC and DIC (line 87). I would suggest to provide an interpretation of the 14C-DIC values without calculating the contribution of mineralised DOC to DIC.

We agree that the low number of DIC age estimates makes interpretations of our mass balance calculations difficult. In line with this comment, we will remove the calculations of the fraction of mineralised DOC contributing to DIC in the revised manuscript.

There are some areas in which the manuscript can be improved:

1. The language can be slightly improved: went on to show (line 48), were always lower (line 221), water stable isotopes (line 210; water does not have isotopes but H and O do), relatively stable between seasons (line 221), here (lines 82, 84 and 87), coarse estimates (line 192), in depletion of modern carbon (line 83).

**We will modify the text as suggested.**

2. Please include information about the type of splines used in the generalised additive model. This is important for reproducibility. Alternatively, the authors can make the code publicly available which I strongly recommend.

**We will add this information.**

3. Include all available data in Figure 5.

We will add the one set of values for site P1 on the figure.